# Guide Star Selection for the Three-FOV Daytime Star Sensor

**DOI:** 10.3390/s19061457

**Published:** 2019-03-25

**Authors:** Liang Wu, Qian Xu, Haojing Wang, Hongwu Lyu, Kaipeng Li

**Affiliations:** 1Department of Computer Science and Engineering, Changchun University of Technology, Changchun 130012, China; xuqianabc@foxmail.com (Q.X.); lvhongwu@mail.ccut.edu.cn (H.L.); likaipeng97@163.com (K.L.); 2Changchun Institute of Optics, Fine Mechanics and Physics, Chinese Academy of Sciences, Changchun 130033, China; wanghaojing@foxmail.com

**Keywords:** star sensor, guide star selection, three-FOV, daytime, SWIR

## Abstract

To realize the application of the star sensor in the all-day carrier platform, a three-field-of-view (three-FOV) star sensor in short-wave infrared (SWIR) band is considered. This new prototype employs new techniques that can improve the detection capability of the star sensor, when the huge size of star identification feature database becomes a big obstacle. Hence, a way to thin the guide star catalog for three-FOV daytime star sensor is studied. Firstly, an introduction of three-FOV star sensor and an example of three-FOV daytime star sensor with narrow FOV are presented. According to this model and the requirement of triangular star identification method, two constraints based on the number and the brightness of the stars in FOV are put forward for guide star selection. Then on the basis of these constraints, the improved spherical spiral method (ISSM) is proposed and the optimal number of reference points of ISSM is discussed. Finally, to demonstrate the performance of the ISSM, guide star catalogs are generated by ISSM, magnitude filter method (MFM), 1st order self-organizing guide star selection method (1st-SOPM) and the spherical spiral method (SSM), respectively. The results show that the guide star catalog generated by ISSM has the smallest size and the number and brightness characteristics of its guide stars are better than the other methods. ISSM is effective for the guide star selection in the three-FOV daytime star sensor.

## 1. Introduction

A star sensor is an optical navigation instrument with high precision, and it has been used in many different applications [1,2,3,4]. One of them is the navigation inside the atmosphere and daytime navigation is one of the most difficult aspects of this kind of application [5]. Because in the visible waveband, intense daytime sunlight scattered in the atmosphere leads to background illumination, making starlight difficult to be detected [6,7,8]. The techniques of multiple fields of view and short-wave infrared (SWIR) are two ways to solve the problem. The multiple fields of view enhance the detection capability of the star sensor through redundant field of view(FOV) [9,10,11]. SWIR is a portion of the electromagnetic spectrum, and its wavelength range is 1μm to 3μm [12]. In this paper, we adopt the typical H-band (1.5–1.8 μm) as the reference waveband for our study. In the SWIR band, there is less atmospheric scattering and higher transmission than the visible waveband [13,14]. Therefore, the probability of detecting stars at daytime with an SWIR sensor is much higher than that with a visible sensor. The 2-Micron All Sky Survey Point Source Catalog (2MASS PSC) is utilized [15] as the reference star catalog in the SWIR band.

Although the multi-FOV SWIR star sensor has the advantage of detection performance in the daytime, the star identification of this kind of star sensors is a hard task. The multi-FOV star identification needs to combine the stars of different FOVs to do star identification, so the identification features contain inside-FOV part and between-FOV part, so the size of the identification feature database gets much larger than the single-FOV star sensor [16,17]. Compared with the traditional single-FOV star sensor, the cost of star identification of the multi-FOV star sensor dramatically increases. Therefore, the guide star catalog is an essential part of the star sensor, some selection criteria should be adopted so that the selected guide star catalog can be used for the daytime application of the multi-FOV star sensor. The construction of guide star catalog directly affects the speed and success probability of star identification, as well as the reliability and availability of star sensors.

The most commonly used method for preparing guide star catalog is the magnitude filter method (MFM). The stars whose magnitude are less than or equal to the magnitude threshold (MT) are selected to form the guide star catalog. This method is simple and can match the performance of the optical system, but the distribution of guide stars is obviously uneven, some parts of the catalog have excessive number of stars, and some parts lack of stars [18]. To solve this problem, several guide star selection methods have been proposed, but most of them are oriented to the single-FOV star sensor in visible waveband, such as Boltzmann entropy method [19], rectangular grid method [20] and so on. Bauer [21] presents a spherical spiral method (SSM) to generate uniformly distributed star catalog. Samaan M.A. [22] tests and validates the performance of SSM. Wang [23] further applies SSM to generate the guide star catalog for SWIR single-FOV star sensor; however, the problem of how to get the suitable number of reference points has not been solved. The self-organizing guide star selection method is proposed by Kim [24]. This method can generate uniformly distributed navigation stars while ensuring that at least *n* stars can be measured inside a specified FOV, but the process is too complicated and the brightness of the guide stars is not considered. An improved self-organizing guide star selection method (1st-SOPM) is proposed by our team [25]. This method is designed for the three-FOV star sensor in the visible waveband and the number of the selected stars is inversely proportional to the size of FOV, so when a narrow FOV is used for high navigation accuracy in the daytime application, the guide star catalog generated by 1st-SOPM is too large. Therefore, the guide star catalog obtained through these methods are not appropriate for this case and the main goal of this paper is to find a guide star selection method to meet the requirement of the high-precision three-FOV daytime star sensor.

The remainder of this paper is organized as follows: the model of three-FOV star sensor is introduced in Section 2 and an example of the three-FOV daytime star sensorwith narrow FOV is presented in this section. In Section 3, the constraints for preparing guide star catalog of the three-FOV star sensor are studied and the improved spherical spiral method is proposed. Besides, the optimal number of the reference points of the spherical spiral method is discussed. In Section 4, the guide star catalog is generated and the performance of the guide star selection methods is analyzed by the comparison with other methods. Finally, some conclusions are drawn.

## 2. Model of the Three-FOV Daytime Star Sensor

### 2.1. Introduction of the Three-FOV Star Sensor

Traditional single-FOV star sensor has only one optical system and the size of FOV is normally designed large enough so as to capture sufficient stars to complete the star identification. The idea of cooperative system of sensors [26] is applied to star sensor system, the multi-FOV star sensor can combine the stars in multiple FOVs to do the star identification, so the detection capability and the reliability of the multi-FOV system is better and the FOV can be designed smaller than the single-FOV system. Small size of FOV can get more accurate star position, which helps to improve measurement accuracy. The three-FOV star sensor is a common structure of the multi-FOV star sensor.

The working principle of the three-FOV star sensor is shown in Figure 1. It consists of three-FOV and each FOV can capture stars on its detector. The centroids of stars in each FOV can be obtained by star image preprocessing and centroid calculation. Then the star identification is conducted with the identification feature database and the centroids. Due to the characteristic of the three-FOV star sensor in combining all the stars in the three FOVs for star identification, the identification database is usually quite large. The identification feature database of the three-FOV star sensor is established based on the guide star catalog, obviously, downscaling the guide star catalog is an efficient way to make the identification feature database smaller and speed up identification. The observed stars in each FOV can be identified and matched with the guide stars in the guide star catalog. The accurate vector information of the observed and guide stars can be used to calculate the position, orientation, or attitude of the star sensor.

### 2.2. An Example of the Three-FOV Daytime Star Sensor with Narrow FOV

In some cases, it is necessary to use high-precision star sensors, even requiring accuracy to reach the sub-arcsecond range, such as precise autonomous geolocation on the surface of the earth. For this purpose, the star sensor needs to be able to compute observed star vectors with uncertainties in the sub-arcsecond range. It is well known that the accuracy of observed star vectors is related to the centroid extraction algorithm, the size of FOV, and the resolution of the star sensor camera. Since the resolutions of current SWIR cameras are usually not higher than 640 × 512, the resolutions are much lower than that of the visible-band cameras. Therefore, with the same centroid extraction algorithm, the high precision of observed star vectors can only be achieved with a narrow FOV.

To better illustrate the guide star selection method in this paper, an example of a high-precision three-FOV daytime star sensor is presented, this example has been inspired by one of the designs given in reference [27]. The parameters of the three-FOV star sensor are as follows. The horizontal projection angle between the boresights of adjacent FOVs is $120^{\circ }$ and the elevation angle is $45^{\circ }$. The size of each FOV is $2.1^{\circ }\times 2.1^{\circ }$, so as to fulfill the attitude accuracy requirements. The principal point is located in the center of the target plane and the optical system has no distortion.

Triangular star identification method is the most widely used and mature method at present, so we adopt this method as the star identification method of the three-FOV star sensor. In line with the requirement of the triangular star identification, the basic constraint of the star number in each FOV can be defined as
(1)$$E=\left\{N_{i}\geq 1,i=0,1,2\right\}$$
where *E* denotes the case that the constraint is satisfied, $N_{i}$ denotes the number of guide stars in each FOV, and *i* denotes the index of the three FOVs: 0, 1 and 2. Equation (Equation 1) implies each FOV need to capture at least one star. To satisfy the constraint, even in the conditions that FOVs pointing to regions with few stars, the limit detection magnitude of the star sensor must be up to magnitude 6.5 in H-band.

This example is used for demonstrating the performance of the guide star selection methods in the following parts.

## 3. The Improved Spherical Spiral Guide Star Selection Method

### 3.1. The Guide-Star-Selection Criteria

With MFM guide star selection, the processed guide star catalog still has a large number of stars and the distribution of stars is uneven. The large guide star catalog leads to a huge identification feature database and it is hard to complete the star identification with this feature database. To solve this problem, we consider the following two constraints for the guide star selection of the three-FOV star sensor.

A. The constraint about the number of stars in FOV 

This constraint mainly guarantees the completeness and local uniformity of the guide star catalog. The basic requirement of triangular star identification is that each time at least three guide stars should be captured. If each FOV of the three-FOV star sensor has one guide star, then the combination of the guide stars in the three FOVs is exactly three guide stars, which can meet the requirements of triangular star identification. Therefore, the first constraint for selecting guide stars (referred to as Selection Constraint *I*) is to reduce the number of guide stars as much as possible under the premise of ensuring that there is one guide star in each FOV. Let the number of guide stars captured in each FOV be ${StarCount}_{i}$. Then the Selection Constraint *I* can be expressed by Formula (1) and the minimum number of stars per FOV is guaranteed.

Notably, this constraint is for the guide star selection. As for the star identification, we can do it in two steps. Firstly, the stars are identified based on the guide star catalog. Then if there are more observed stars than the guide stars in the FOVs, we can continue to identify the rest observed stars with the original star catalog according to the identified stars. With the known stars, the second step of the identification can be much faster and easier. Meanwhile, in this way, the identified stars in the first step can be checked.

B. The constraint about the optimal brightness 

This constraint mainly guarantees the optimal brightness. According to the Selection Constraint *I*, the star catalog should be kept as small as possible under the premise that there is at least one guide star in each FOV; however usually there are much more stars in the FOV than the constraint. How can we select the guide stars from so many candidate stars? This constraint can solve the problem.

In this paper, according to the star images captured by the prototype of the star sensor, the relationship among star magnitude, gray level, and the background simulation parameters is analyzed. Then, according to the relationship, the stars of 3 to 9 magnitude are synthesized into an image with an equal interval of 0.5 magnitudes, as shown in Figure 2. In the simulated star image, the nominal position ($x_{t},y_{t}$) of each star is known.

The centroid positions ($x_{s},y_{s}$) of the stars with different magnitudes are obtained by weighted centroid method. The estimation errors of the centroids are calculated as
(2)$$PosErr=\sqrt{{\left(x_{s}-x_{t}\right)}^{2}+{\left(y_{s}-y_{t}\right)}^{2}}$$
The relationship between the centroid estimation error and the magnitude of stars is shown in Figure 3. It can be seen that the centroid estimation accuracy is inversely proportional to the magnitude of stars. The reason is that the star with higher brightness usually has higher SNR and the smaller centroid estimation error is obtained.

There are two advantages of selecting the brightest star in the FOV as guide star. Firstly, as the analysis above, with the brightest star, we can get a more accurate star position. Secondly, we can choose the brightest star in the FOV to conduct the star identification, so that the procedure can be done in less time. Hence, the brightest star in the FOV should be chosen as the guide star, and this is the second constraint of guide star selection (referred to as Selection Constraint II).

It should be pointed out that if the brightest observed star in the FOV is not a guide star, the second or third brightest observed star should be taken into the star identification. This situation may increase the time of star identification, but it will not influence the results. The Selection Constraint II is an efficient way to reduce the probability of that happening.

Based on the two constraints mentioned above, the guide star selection method for the three-FOV daytime star sensor is established to reduce the size of the guide star catalog as much as possible.

### 3.2. The Improved Spherical Spiral Methods

In our previous work, an improved self-organizing guide star selection method for three-FOV star sensor in the visible waveband has been proposed. The Tycho-2 catalog has been used as the basic star catalog and the simplified guide star catalog has achieved good performance. However, for the SWIR band, the previous method leads to a big size of the guide star catalog for the small FOV. For example, we use the FOV with a radius of $1^{\circ }$ and the size of the guide star catalog is 34,633. The size of the guide feature database corresponding to this number of guide stars is too large to complete the star identification. In this part, we propose two improved methods based on the spherical spiral guide star selection method to get a smaller guide star catalog which is suitable for the SWIR band.

#### 3.2.1. Introduction of the Spherical Spiral Method

In 2000, Robert Bauer proposed the spherical spiral method (referred to as SSM) [21]. With this method, *N* evenly distributed rotating reference points on the spherical surface can be obtained. The coordinates of these reference points can be calculated by Formulas (3)–(5).
(3)$$L=\sqrt{N\pi }$$
(4)$$\left\{\begin{array}{c} z_{i}=1-\frac{2i+1}{N}\\ \phi_{i}=\arccos z_{i}\\ \theta_{i}=L\phi_{i} \end{array}\right.$$
(5)$$\left\{\begin{array}{c} x_{i}=\sin\phi_{i}\cos\theta_{i}\\ y_{i}=\sin\phi_{i}\sin\theta_{i} \end{array}\right.$$
where *N* is the number of reference points to be constructed, ($x_{i},y_{i},z_{i}$) is the position of the *i*th spired reference point in the Cartesian coordinate.

The illustration of SSM is shown in Figure 4. Guide stars are obtained from the reference spherical circles centered by the reference points, and the stars which are close to the reference points are selected as guide stars. The main idea of SSM is to generate sufficient number of sampling boresight directions on the celestial sphere so that these sampling reference spherical circles can represent the most situations of the FOV of the star sensor.

The advantage of this method is to guarantee the completeness and local uniformity of the guide star catalog, and with suitable value of *N*, the guide star catalog can meet Selection Constraint *I*. However, the method does not consider the brightness of the guide stars, so it is not in accordance with Selection Constraint II and the guide star catalog which is constructed directly by this method is not applicable to the three-FOV star identification. Furthermore, at present, the way to determine the suitable value of *N* is not presented.

#### 3.2.2. Improved Spherical Spiral Method Based on Selection Constraint II


On the basis of SSM, we propose two improved methods to keep the guide star catalog following Selection Constraint II.

A. The improved spherical spiral method 1(ISSM1) 

SSM chooses the stars close to the center of the FOV as the guide stars, but the improved method selects the brightest star in the FOV as the guide star in order to meet Selection Constraint II. The specific steps are as follows:(1)Generate the basic guide star catalog from the 2Mass PSC star catalog with MFM method. The magnitude of MFM method is based on the limit detection magnitude of the star sensor.(2)Let *i* = 0.(3)Calculate the *i*th position of the reference point with Equations (3)–(5).(4)Determine the *i*th reference spherical circle. The center of the reference spherical circle is the position of the reference point. In addition, the radius of the reference spherical circle is less than or equal to the radius of the FOV of the star sensor.(5)Extract the set of candidate guide stars $S_{i}$ in the *i*th reference spherical circle from the basic guide star catalog.(6)If there is only one star in $S_{i}$, the star is marked as the guide star directly. Otherwise, the stars in $S_{i}$ should be sorted according to the magnitude, and the brightest star is marked.(7)Let *i* = *i*+1.(8)Repeat step (3) to (7) until *i* = *N*. *N* is the total number of the reference points.(9)Construct guide star catalog with all the marked stars.

B. The improved spherical spiral method 2 (ISSM2) 

To further reduce the number of guide stars, we improve step (6) of ISSM1. In this improved method, if more than one of the candidate stars in $S_{i}$ has been marked as guide stars, then we don’t need to sort the candidate stars by magnitude and the algorithm jumps to step (7). Otherwise, if there is not any marked guide star in $S_{i}$, the procedure is just like step (6) of ISSM1. Compared ISSM2 with ISSM1, it is obvious that the number of the guide stars is reduced; however, this comes at the expense of the Selection Constraint II because the selected guide star maybe not the brightest candidate star in the reference spherical circle.

The procedure of ISSM1 and ISSM2 are represented in Figure 5. The steps with solid line belong to both ISSM1 and ISSM2, and the steps with dotted line are the modification from ISSM1 to ISSM2.

The number of guide stars obtained by different methods and different number of reference points is shown in Figure 6. The number of guide stars with SSM is much more than the two improved methods, and the selected guide stars with ISSM2 is lesser than ISSM1, which is consistent with our expectations. With the increase of reference points, the number of guide stars shows a steady increasing trend and the completeness of the guide star catalog is better. However, the burden of star identification is also increased. So, the way to find an optimal number of the reference points is discussed in Section 3.2.3.

#### 3.2.3. Study on the Optimal Number of Reference Points

The probability of FOV without guide star has been tested. The guide star catalogs are generated by SSM, ISSM1 and ISSM2 with different number of reference points. The test is performed in the platform which will be described in detail in Section 4.2. The result is shown in Figure 7. With the growth of the number of reference points, all the three methods can achieve very good completeness. Moreover, the numbers of reference points used by the three methods to achieve complete guide star catalog are consistent, approximately 50,000 reference points. This consistency leads us to think about whether there is an optimal number of reference points for such methods.

The coverage of the reference spherical circle is analyzed. The surface area of a sphere is:(6)$$S_{0}=4\pi r^{2}$$
here we can treat *r* as 1 or an arbitrary value. For a circular FOV, the area of one reference spherical circle can be calculated by
(7)$$S_{FOV}=2\pi r^{2}\left(1-\cos FOV_{r}\right)$$
where $FOV_{r}$ is the radius of the FOV.

Thus, the number of reference spherical circles for covering the whole sphere is:(8)$$N_{0}=\frac{S_{0}}{S_{FOV}}=\frac{2}{1-\cos FOV_{r}}$$

For example, the radius of the FOV is $1^{\circ }$, by Formula (8), 13,132 reference spherical circles can cover the whole sphere. According to Figure 7, the least number of reference points for satisfying the completeness is 50,000, which is about four times of $N_{0}$. Figure 8 shows the coverage of the spherical circles on the sphere when the number of reference points is 1, 2, 3, 4 times of $N_{0}$. The black circle is an arbitrary reference spherical circle and the green circles are the adjacent spherical circles. It can be seen that when the number of reference points is 4 times of $N_{0}$, the black circle is overlapped by the adjacent reference spherical circles. The completeness of the guide star catalog is consistent with the coverage area of reference spherical circles and it is appropriate to select 4 times of $N_{0}$ as the number of reference points.

To verify this assumption, we use different radiuses of FOV and different multiples of $N_{0}$ as the number of reference points to generate the guide star catalog with ISSM2. Then the Monte Carlo test as described in Section 4.2 is conducted and the results are shown in Figure 9.

In Figure 9, the *x*-axis represents the ratio between the number of the reference points and $N_{0}$, and the *y*-axis represents the probability without guide star in the FOV. It can be seen from the figure that when the number of reference points is more than four times of $N_{0}$, an almost complete catalog can be constructed. The results of different size of FOV are consistent. Therefore, 4 times of $N_{0}$ is a suitable number of reference points to keep the minimum size and completeness of the guide star catalog.

## 4. Results and Discussion

### 4.1. Characteristics of Guide Star Catalogs Generated by Different Methods

The parameters of the star sensor are listed in Section 2.2. We choose the 2Mass PSC as the basic star catalog and use different methods to generate guide star catalogs according to this star sensor model. The numbers of the guide stars in each catalog are listed in Table 1.

The method of MFM 6.5 represents MFM method with MT 6.5. 1st-SOPM method is the guide star selection method for the three-FOV star sensor in visible waveband which is used in our previous work. According to the analysis of Section 3.2.3, SSM, ISSM1, and ISSM2 are conducted with the 4 times of $N_{0}$ as the number of reference points. Among the methods, the guide star catalogs based on ISSM1 and ISSM2 have the minimum sizes, and the sizes are reduced by 87.70% and 89.27% relatively compared with MFM method, 31.75% and 40.48% respectively compared with 1st-SOPM.

The magnitude characteristics of the guide star catalogs are shown in Figure 10. The magnitude distribution of guide star catalog generated by SSM (Figure 10b) is basically consistent with the one generated by using MFM method directly (Figure 10a), so SSM is not in accordance with Selection Constraint II, this result is the same with the analysis in Section 3.2.1. The magnitude characteristic of 1st-SOPM method (Figure 10c) is better than SSM, but the magnitudes of the guide stars are still concentrated in the range of 4–6 magnitude. The magnitude characteristics of ISSM1 (Figure 10d) and ISSM2 (Figure 10e) are inclined to the low-magnitude region compared with the 1st-SOPM method, so, ISSM1 and ISSM2 are more in line with Selection Constraint II. The magnitude distribution of the guide star catalog generated by ISSM2 is not as good as ISSM1, which is because not all the brightest candidate stars in reference spherical circle are chosen as guide stars in ISSM2.

### 4.2. Performance of the Guide Star Catalogs Generated by Different Methods

A test platform based on the Monte Carlo method is established for analyzing the performance of the guide star catalogs generated by different methods. The flow chart is shown in Figure 11. For each simulation, a random boresight direction of FOV is selected. Then the star set $S_{1}$ of the FOV is extracted from the basic star catalog which is generated by MFM 6.5 method. Meanwhile, the guide star set $S_{2}$ is extracted from the guide star catalog. $N_{1}$ and $N_{2}$ are the numbers of the stars in $S_{1}$ and $S_{2}$ respectively. If $N_{1}$ is equal to 0, which means that in this case, there is no observed star in the FOV, so we treat this case as invalid case. Otherwise, the number of guide stars $N_{2}$ is recorded for the following analysis. Besides, the brightest guide star *S* is extracted from the star set $S_{2}$ and the rank of brightness of *S* in the star set $S_{1}$ is recorded. By this index, whether the guide star catalog is in line with Selection Constraint II can be analyzed. The number of Monte Carlo simulations is 10,000. As shown by the analysis performed in Section 4.1, SSM does not follow Selection Constraint II, so SSM is discarded in the following analysis.

With the simulation result $N_{2}$, the performance related to the Selection Constraint *I* is analyzed. For each guide star selection method, the probability of the guide star number in FOV can be calculated with $N_{2}$, and the probabilities of ISSM1, ISSM2 and 1st-SOPM are shown in Figure 12. All the three methods can meet the requirements of Selection Constraint *I*. There are guide stars in every FOV. Among them, the mean value of guide stars in the FOV of ISSM1 is 3.618573, and the standard deviation is 1.172465. For ISSM2, the mean value is 3.151795 and the standard deviation is 1.029260. For 1st-SOPM, the mean value is 5.297090 and the standard deviation is 1.393487. It can be seen that the number of guide stars in the FOV of ISSM2 is the least and the local uniformity is the best. Therefore, with regard to Selection Constraint *I*, the performance of ISSM2 is the best among the methods, and ISSM1 is better than 1st-SOPM.

With the rank of brightness of the guide star, the performance related to the Selection Constraint II is analyzed. The probability of brightness of the guide stars in FOV is counted from the simulation results, and the probability is listed in Table 2. The result shows that the probability of the case that the brightest observed star is the guide star with ISSM1, ISSM2 and 1st-SOPM are 98.3%, 89.32%, and 90.43% respectively. As for the performance related to Selection Constraint II, ISSM1 is obviously better than the other two methods. The brightness performance of ISSM1 is better than that of ISSM2 because ISSM1 acquires more guide stars and chooses the brightest stars in each reference spherical circle. The performance of ISSM2 and 1st-SOPM are similar, but the size of guide star catalog of ISSM2 is far smaller than that of 1st-SOPM.

In summary, ISSM1 and ISSM2 meet well the requirements of the two selection criteria of the three-FOV daytime star sensor. As for the Selection Constraint *I*, the performance of ISSM2 is better than ISSM1, whereas as for the Selection Constraint II, the contrary is the case. On the whole, for the application of SWIR band, the performance of both ISSM1 and ISSM2 are better than 1st-SOPM, and the two methods are quite similar.

Here we imagine the different application of ISSM1 and ISSM2 based on their characteristics. The first case is the all-sky star identification (star identification without prior information), in this case, the guide star catalog size should be made as small as possible, so ISSM2 can be chosen to minimize the number of guide stars so that the size of navigation feature database can be reduced and the process of star identification is sped up. The second case is local recognition, the navigation feature database can be greatly reduced according to prior information. At this time, the influence of the number of guide stars on star identification is relatively weaker, and the brightest star in the FOV is the guide star, so with the guide star catalog generated by ISSM1, we can directly use the brightest star of each FOV to do the star identification and the efficiency of star identification is improved.

## 5. Conclusions

To improve the efficiency of star identification for the three-FOV daytime star sensor, two new guide star selection methods are proposed. First of all, an examplea three-FOV star sensor is constructed in the SWIR band to meet the requirement of the high-precision daytime application, and the 2MASS PSC is selected as the original catalog. Then two selection criteria are put forward and based on these criteria two improved spherical spiral methods are proposed and the optimal number of the reference points is studied. When the number of reference points is 4 times of the number of FOVs which can cover the whole celestial sphere, the completeness of the guide star catalog can be satisfied. With the proper number of reference points, the guide star catalogs generated by ISSM1 and ISSM2 are 87.70% and 89.27% smaller than the basic star catalog, respectively. Finally, we use the Monte Carlo simulation platform to test the performance of different guide star selection methods. Compared with our previous work, 1st-SOPM, the local uniformity of ISSM1 and ISSM2 are better and there are fewer guide stars inside the FOV, and the star identification can be sped up theoretically. Meanwhile, the brightness characteristic of ISSM1 and ISSM2 are better than 1st-SOPM. In summary, the star catalog generation method described in this paper provides a better performance for the daytime application of the three-FOV star sensor. Moreover, these methods are also appropriate for multiple-FOV star sensors operating in other optical wavebands.

## Figures and Tables

**Figure 1 sensors-19-01457-f001:**
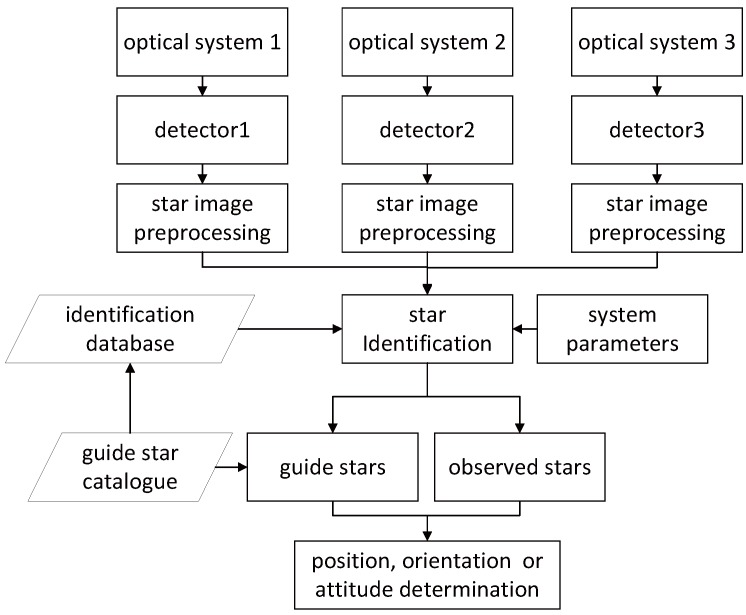
Working principle of the three-FOV star sensor.

**Figure 2 sensors-19-01457-f002:**
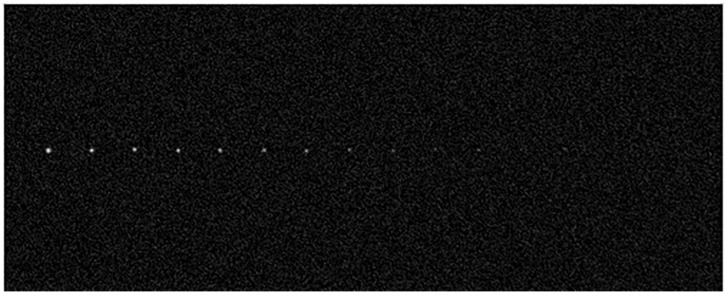
A simulated star image based on the parameters of the star.

**Figure 3 sensors-19-01457-f003:**
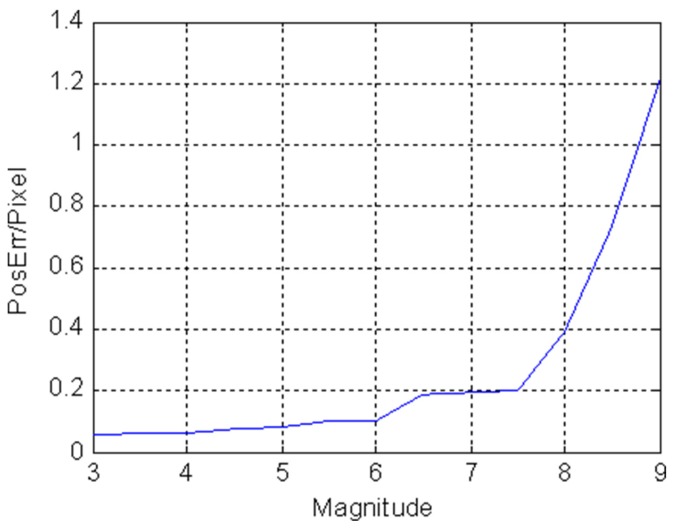
The relationship between centroid estimation error and magnitude.

**Figure 4 sensors-19-01457-f004:**
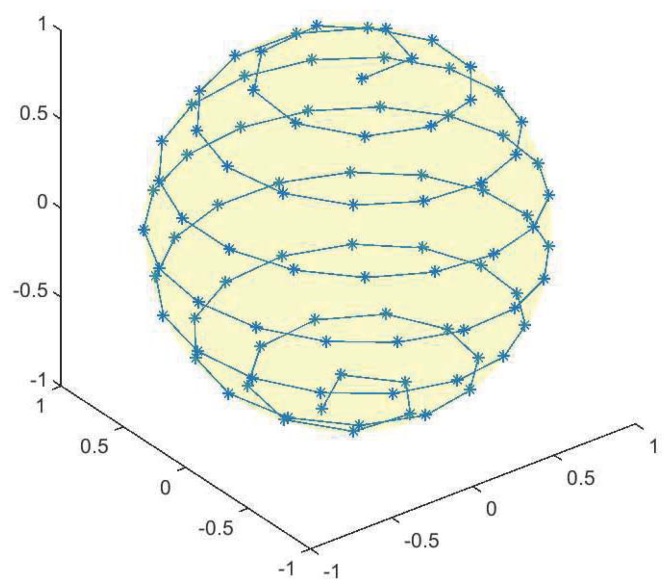
The illustration of SSM.

**Figure 5 sensors-19-01457-f005:**
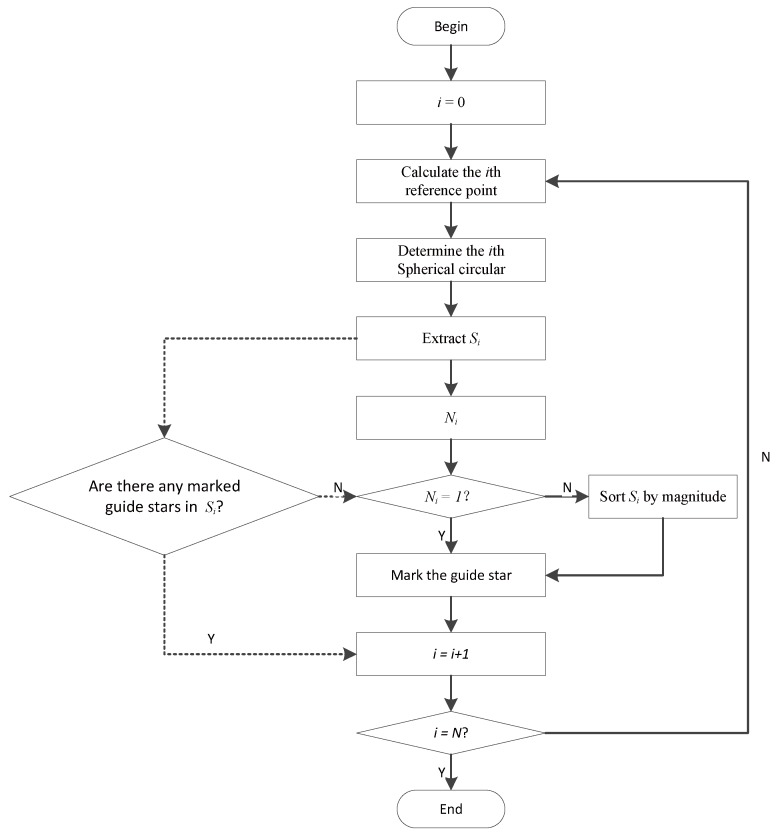
Flow chart of ISSM1 and ISSM2.

**Figure 6 sensors-19-01457-f006:**
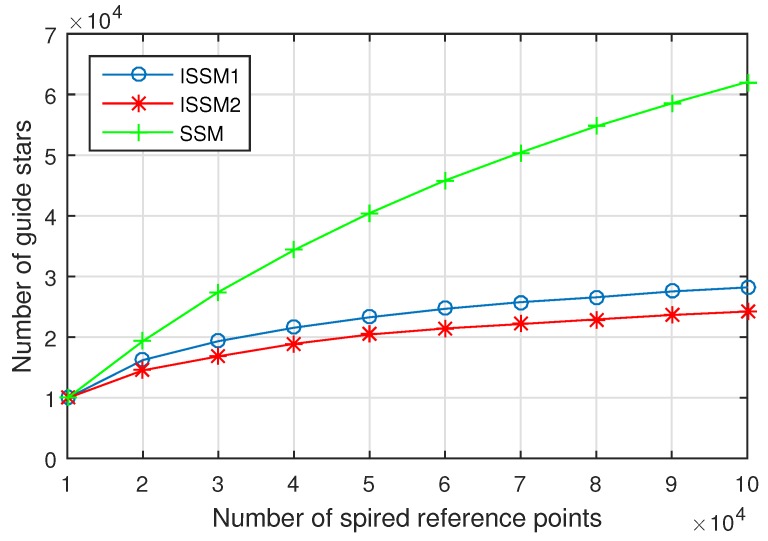
The relation between the number of guide stars and the number of reference points in FOV.

**Figure 7 sensors-19-01457-f007:**
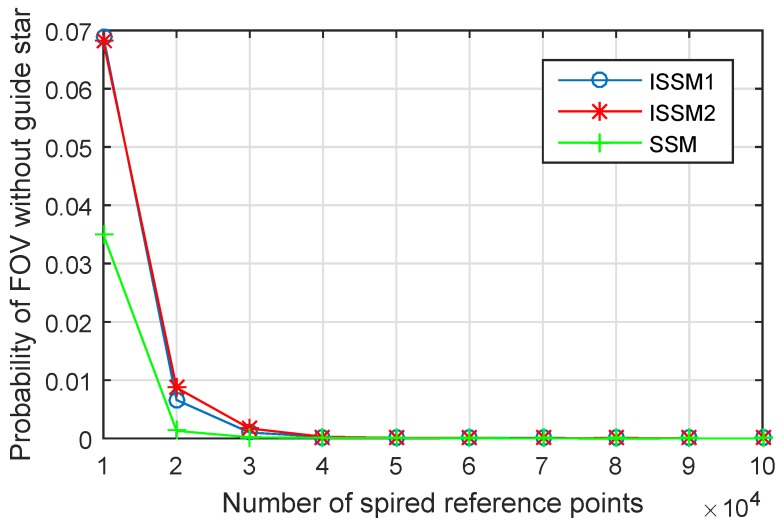
Probability of the FOV without guide star.

**Figure 8 sensors-19-01457-f008:**
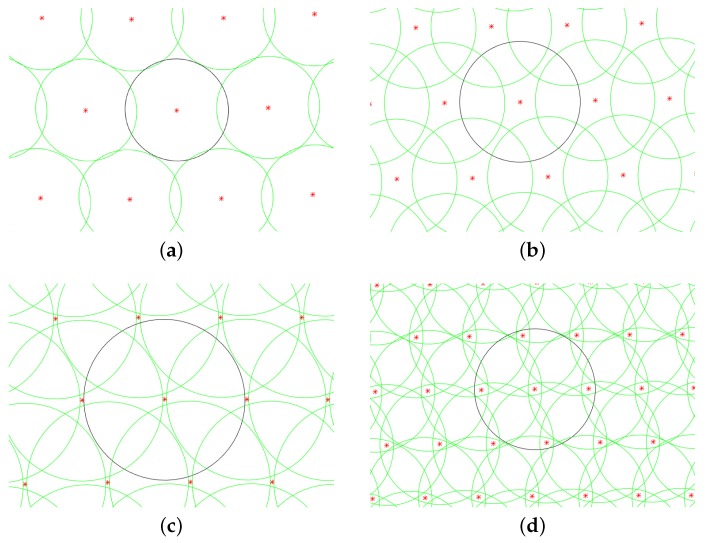
Spherical coverage of reference FOV. (**a**–**d**) are respectively the covering states of the reference FOV on the sphere when the number of reference points is 1, 2, 3 and 4 times of $N_{0}$.

**Figure 9 sensors-19-01457-f009:**
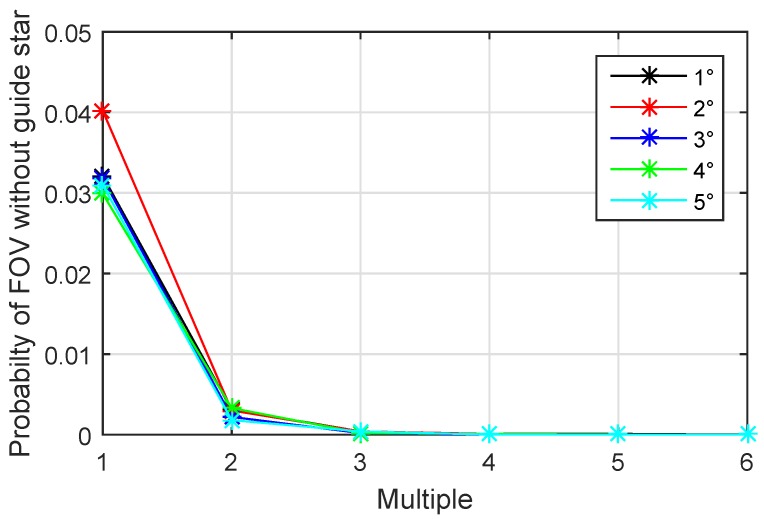
Probability without guide star in different FOV sizes.

**Figure 10 sensors-19-01457-f010:**
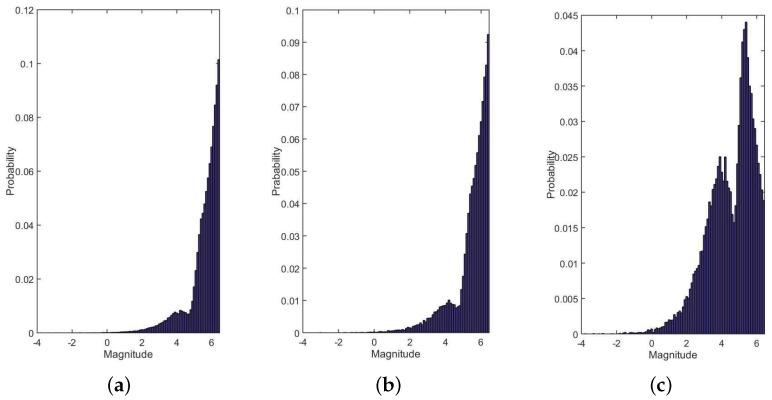
Magnitude arrangement of guide catalogs based on different methods (the width of bins used in the histograms is 0.1 magnitude). (**a**) MFM 6.5; (**b**) SSM; (**c**) 1st-SOPM; (**d**) ISSM1; (**e**) ISSM2.

**Figure 11 sensors-19-01457-f011:**
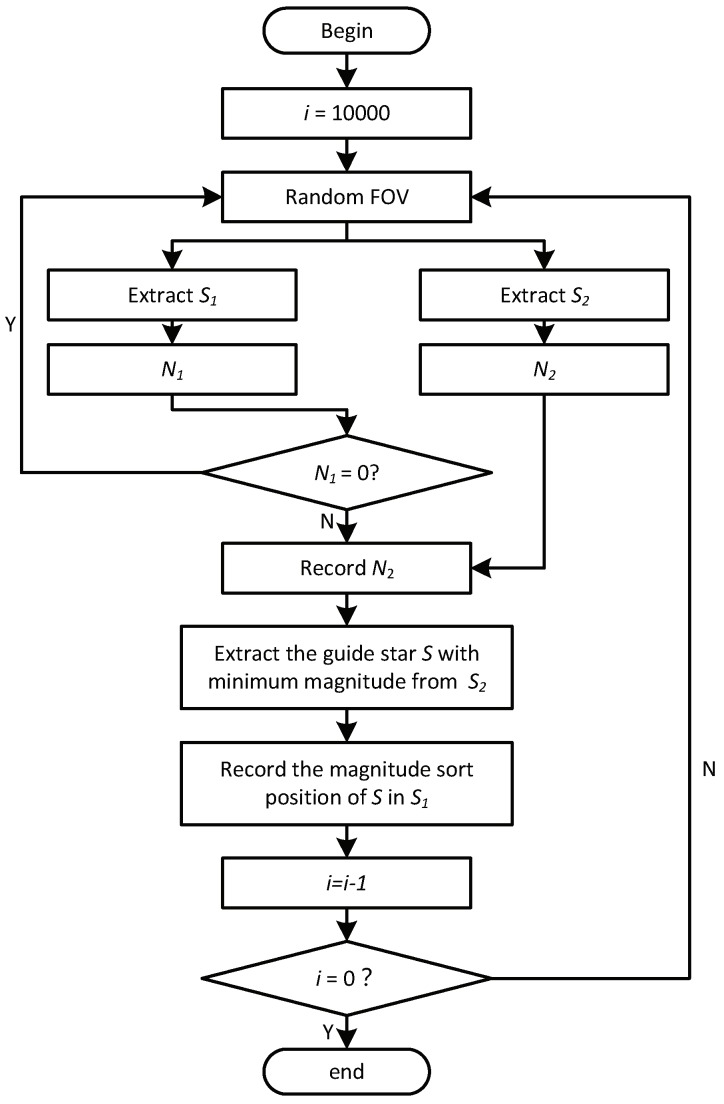
Flow chart of the performance test.

**Figure 12 sensors-19-01457-f012:**
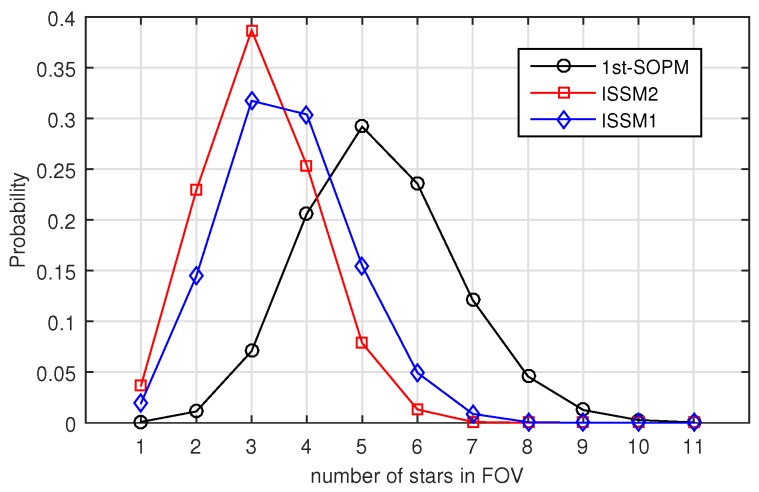
Probability of the number of guide stars in FOV with different algorithms.

**Table 1 sensors-19-01457-t001:** The guide star catalogs based on different methods.

Method	MFM 6.5	1st-SOPM	SSM	ISSM1	ISSM2
Number of guide stars	192,181	34,633	41,561	23,638	20,615

**Table 2 sensors-19-01457-t002:** Probability statistics of brightness of the guide stars in FOV.

Method	Probability of Brightness Sort for Guide Star in FOV (%)					
	0	1	2	3	4	5
ISSM1	98.30	1.45	0.18	0.04	0.01	0.01
ISSM2	89.32	8.41	1.56	0.44	0.13	0.07
1st-SOPM	90.43	8.10	1.17	0.24	0.04	0.01

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
