# Peer review of "Guide Star Selection for the Three-FOV Daytime Star Sensor"

_sensors, 2019, doi:10.3390/s19061457_

Round 1
Reviewer 1 Report
Congratulations for the good job. The paper is promising, but many items need to be addressed. Please see comments in the attached file.

Author Response
First, we would like to extend our appreciation for the reviewer’s comments concerning our manuscript entitled “Guide Star Selection for the three-FOV Daytime Star Sensor”. The comments are the important guiding significance to our further researches. We have studied comments carefully and have made revisions which we hope meet with approval. Revised portion are marked in the paper. All the lines indicated below are also in the revised version. Please see the attached file for the main corrections in the paper and the response to the reviewer's comments.

Reviewer 2 Report
The paper presents new results that could be considered correct.
In the complex area of star mapping it is also useful to consider network of sensors and alternative strategies for data handling. It is useful to consider networks based on self organization and cooperative systems. I suggest for future studies to take into account these concepts. Moreover I suggest in this view to include in the paper the following reference:
Annals of GeophysicsOpen AccessVolume 54, Issue 5, 2011, Pages 544-550
The HOTSAT volcano monitoring system based on combined use of SEVIRI and MODIS multispectral data(Article)
Ganci, G.a,
Vicari, A.a,
Fortuna, L.b,
del Negro, C.a
Even if the paper is referred to a different kind of research it can be useful to explain the otlined previously ideas.
The paper presents new results that could be considered correct.
Inthe complex area of star mapping it is also useful to consider network of sensors and alternative strategies for data handling. It is useful to consider networks based on self organization and cooperative systems. I suggest for future studies to take into account these concepts. Moreover I suggest in this view to include in the paper the following reference:
Ganci, G.a,
Vicari, A.a,
Fortuna, L.b,
del Negro, C.a
Author Response

(The authors gave the same response as above.)

Round 2
Reviewer 1 Report
The authors did a great job in reviewing the paper. However, some points still need to be addressed.
A better justification for the selection of the three FOV star sensor with narrow FOV is needed.

Author Response

(The authors gave the same response as above.)
